# The impact of specialised gastroenterology services for pelvic radiation disease (PRD): Results from the prospective multi-centre EAGLE study

John N. Staffurth[1,2‡], Stephanie Sivell [3‡*], Elin Baddeley[3], Sam Ahmedzai[4], H. Jervoise Andreyev[5,6], Susan Campbell[3,7], Damian J. J. Farnell[8], Catherine Ferguson[9], John Green[10], Ann Muls[11], Raymond O'Shea[3], Sara Pickett[12], Lesley Smith[13], Sophia Taylor [14], Annmarie Nelson[3]

1 Division of Cancer and Genetics, School of Medicine, Cardiff University, Cardiff, United Kingdom, 2 Velindre Cancer Centre, Cardiff, United Kingdom, 3 Marie Curie Research Centre, Division of Population Medicine, School of Medicine, Cardiff University, Cardiff, United Kingdom, 4 National Institute for Health Research, Clinical Research Network–Cancer Cluster Office, University of Leeds, Leeds, United Kingdom, 5 Department of Gastroenterology, Lincoln County Hospital, Lincoln, United Kingdom, 6 The School of Medicine, University of Nottingham, Nottingham, United Kingdom, 7 Wales Cancer Research Centre, Cardiff, United Kingdom, 8 School of Dentistry, Cardiff University, Cardiff, United Kingdom, 9 University of Sheffield Medical School, Sheffield, United Kingdom, 10 Department of Gastroenterology, Cardiff and Vale University Health Board, Cardiff, United Kingdom, 11 Royal Marsden NHS Foundation Trust, London, United Kingdom, 12 Swansea Centre for Health Economics, Swansea University, Swansea, United Kingdom, 13 Living With and Beyond Cancer Programme, NHS England, London, United Kingdom, 14 School of Health Sciences, University of Southampton, Southampton, United Kingdom

‡ JNS and SS are joint first authors on this work.
* SivellS2@cardiff.ac.uk

**Data Availability Statement:** All relevant data are within the manuscript and its Supporting Information files.

## Abstract

To undertake a mixed-methodology implementation study to improve the well-being of men with gastrointestinal late effects following radical radiotherapy for prostate cancer. All men completed a validated screening tool for late bowel effects (ALERT-B) and the Gastrointestinal Symptom Rating Score (GSRS); men with a positive score on ALERT-B were offered management following a peer reviewed algorithm for pelvic radiation disease (PRD). Health-related quality of life (HRQoL) at baseline, 6 and 12 months; and healthcare resource usage (HRU) and patient, support-giver, staff experience and acceptability of staff training (qualitative analysis) were assessed. Two nurse- and one doctor-led gastroenterology services were set up in three UK cancer centres. Men (n = 339) who had had radical radiotherapy for prostate cancer at least 6 months previously, were recruited; of which 91/339 were eligible to participate; 58/91 men (63.7%) accepted the referral. Diagnoses included: radiation proctopathy (n = 18); bile acid malabsorption (n = 15); fructose or lactose intolerance and/or small intestinal bacterial overgrowth (n = 20); vitamin B12/D deficiency (n = 20). Increases in quality of life, sexual activity and/or sexual function, and decrease in specific symptoms (e.g. bowel-related or urinary) between 6 and 12 months were observed. Limited HRU modelling suggested staff costs were £117-£185, depending on the service model; total costs averaged £2,243 per patient. Both staff and patients welcomed the new service although there was concern about long-term funding and sustainability beyond the

**Funding:** This study was funded by Prostate Cancer UK's TrueNTH initiative (Grant Reference No. 250-55).

**Competing interests:** The authors have declared that no competing interests exist.

timeframe of the study (qualitative). PRD is increasingly recognised worldwide as an ongoing consequence of curative pelvic radiotherapy, despite widespread implementation of advanced radiotherapy techniques. Specialised services following national guidelines are required.

## Introduction

Pelvic radiotherapy is widely used in prostate cancer and other malignancies [1–3]. Acute side effects often resolve within a few months [4]. Late radiation effects after pelvic radiotherapy, also termed pelvic radiation disease (PRD) [5]; occur after 3 months and are often considered irreversible and an accepted consequence of treatment for prostate cancer [4]. They can occur many years after the radiotherapy and can have a significant impact on quality of life [6], but treatment and support can alleviate the severity of these symptoms. We report the main findings from the EAGLE study (*Improving the Wellbeing of Men by Evaluating and Addressing the Gastrointestinal Late Effects of Radical Treatment for Prostate Cancer*)—a mixed-methodology implementation study.

PRD can affect multiple domains: gastrointestinal, genitourinary, sexual, endocrine, skeletal, psychological and fatigue. While there is some uncertainty over the true prevalence of late effects [7, 8]. It is important to recognise that these effects on different domains may co-exist in the same patient, and inked and can impact on a man's psychological and social wellbeing, including feelings of regret at their choice of cancer treatment [9–12]. Oncologists often do not seek information on late effects focusing on symptoms of cancer recurrence [4]. Patients, viewing them as expected consequences of therapy, may be too embarrassed to mention them, or grateful for being cured, may not mention their symptoms [13]. Many different reporting systems and tools are used to collect toxicity and quality of life data in both research and routine practice [7, 8, 14]. There is no agreed definition of pelvic radiation disease and its prevalence depends on radiation delivered (dose, volume and fractionation), patient factors and follow-up, including extent of investigations for symptoms consistent with PRD [15]. Therefore, the prevalence of grade two or higher late gastrointestinal and genitourinary effects after modern radiotherapy for localised prostate cancer is usually <3% [16], whereas the prevalence of grade 3 or higher toxicity maybe as high as 10% after wide-field radiotherapy for cervical cancer or in the peri-operative setting for rectal cancer [17, 18] prevalence studies may report as high as 40% of patients with long-term bowel dysfunction after such wide-field radiotherapy [19].

Acute radiation effects are caused by damage to rapidly proliferating tissues such as gut mucosa [20]. The pathophysiology of PRD is more complicated and symptoms may be caused by structural or functional changes. Some patients will have a persistent unchecked inflammatory response with an initial acute inflammatory response complicated by endothelial cell and microvascular damage; this sustains tissue hypoxia, dysregulates wound healing and can lead to fibrosis. Hypoxia leads to abnormal neovascularisation i.e. telangiectasia. Direct damage can also occur to larger vessels and nerves; often in tissues that have prior surgery as part of the initial cancer management. Fibrosis within the gastrointestinal tract can alter transit time, impair motility, affect nutrient absorption, alter the microbiome and cause strictures, infections and even fistulae [19]. These physiological changes may lead to over 20 different gastrointestinal or nutritional diagnoses and 12 different symptom [21]. Equivalent changes may affect other domains, although are less well studied [22].

Currently follow-up of men following prostate cancer radiotherapy in the UK is non-standardised and even if late gastrointestinal (GI) side effects are identified [20] there is often no agreed pathway for onwards referral and management [23, 24]. The EAGLE study focused on GI effects of radiotherapy and aimed to enable centres to rapidly adopt best practice in managing GI consequences of treatment, and develop into a series of sustainable centres of excellence.

## Materials and methods

Ethical approval was received from the NHS Research Ethics Committee: NRES Committee North West-Liverpool East REC (Reference 14/NW/1206).

We used the Normalisation Process Theory (NPT) as the overarching framework to collect the data [25, 26]. The outline of the materials and methods are presented below [24].

We aimed to introduce and evaluate innovative services in three UK centres managing men with prostate cancer in the post-curative treatment setting. Our objectives were to:

- Raise awareness and increase expert capacity for the treatment of GI late effects following radiotherapy;

- Implement an interventional model of care to develop centres of excellence, specifically focused on early identification and management of GI late effects;

- Evaluate the effectiveness of the new service via quality of life, healthcare utilisation analysis, health economics, and by assessing the acceptability to healthcare professionals (HCP) and patients/families;

- Monitor the experiences of patients, support-givers and members of staff regarding service implementation.

We used *ALERT-B*, a novel screening tool for late bowel effects following pelvic radiotherapy; it has a single threshold, any positive response will trigger referral for symptoms investigation (see **Fig 1**) [27], and was successfully face tested for acceptability and psychometrically validated against the Gastrointestinal Symptom Rating Score (GSRS) [28].

Enhanced GI Services were developed in 3 'implementation centres'; sites were eligible to participate had they met specific criteria, including: a clinical oncology team which managed at least 100 patients with radiotherapy for prostate cancer per annum; a prostate cancer oncologist willing to use the ALERT-B Tool to identify and recruit patients to the study; a gastroenterologist who could support the introduction of a nurse-led post to adhere to the Royal Marsden Hospital (RMH) algorithm [29], with dietician support [24]. Levels of the service were assessed during the first six months of the study at each of the study sites, confirmed the likely sustainability of each site and assessed any practical solutions to intervention re-design and finalisation. Following this, staff were then recruited and trained to use and deliver the intervention. The training was aimed at both the specialist PRD team and wider non-specialist oncological and gastroenterological teams and comprised the Macmillan Cancer Support-RMH online training module and the GI and nutrition teams at the RMH [24]. Men who had received radical radiotherapy for prostate cancer at least six months previously completed the ALERT-B tool and the Gastrointestinal Symptom Rating Score (GSRS) and were referred to the Enhanced GI Service implementation centre if their scores were positive. Fig 2 shows the desired impact of the specialized gastroenterology service including optimal referral and management pathways and expected outputs including education amongst healthcare professionals, establishing wider service networks and evaluation and research.

---

**Assessment of Late Effects of RadioTherapy-Bowel**
**ALERT-B** Screening Tool

Date:

Your specialist has asked you to complete this screening tool to pick up any bowel or tummy problems you may have developed following radiotherapy treatment.

**Please answer Yes or No to the following questions:**

1. Do you have difficulty in controlling your bowels (having a poo), such as:

   - Having to get up at night to poo Yes ☐ No ☐

   - Having accidents, such as soiling or a Yes ☐ No ☐
     sensation of wetness ("wet wind")

2. Have you noticed any blood from your bottom recently? Yes ☐ No ☐
   (any amount or frequency)

3. Do you have any bowel or tummy problems that affect Yes ☐ No ☐
   your mood, social life, relationships or any other aspect
   of your daily life?
   (e.g., do you avoid any activities or situations- travel, work, social life or hobbies? Do you take
   continence supplies or spare clothing with you when you go out? Have you made any dietary changes?
   Do you need to allow for frequency or urgency of needing the toilet?)

**If you have any other problems your doctor will be happy to discuss this with you.**

**Fig 1. Final version of ALERT-B screening tool questions [18].**

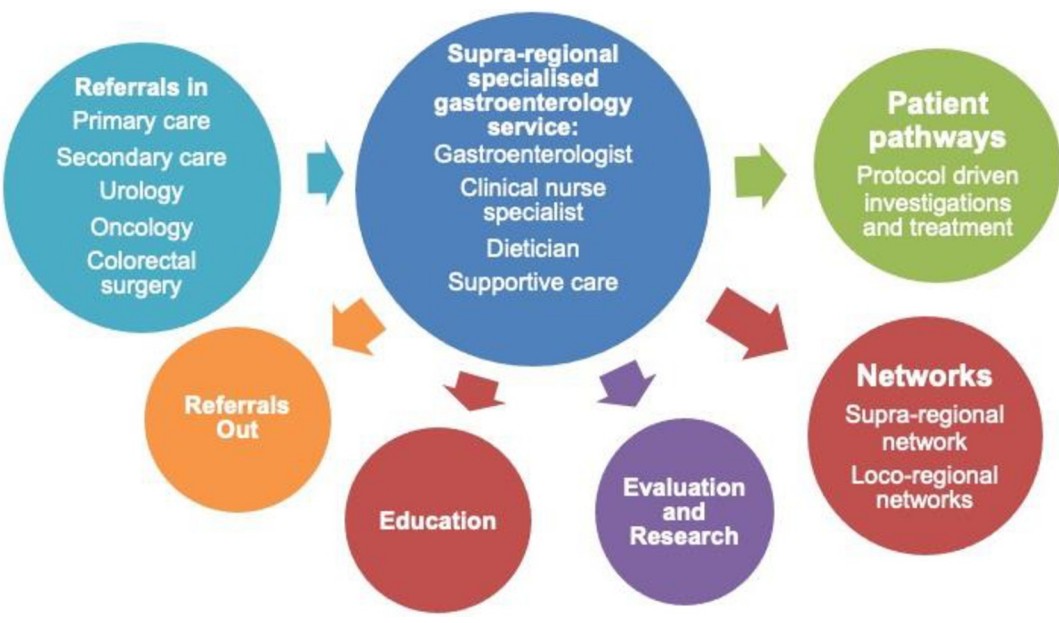

**Fig 2. Desired impact of the new specialised gastroenterology service.**

---

## Participants

All patients attending specific uro-oncology follow up clinics were screened. Inclusion criteria for the main EAGLE study, including the qualitative interviews, were as follows: Patients attending uro-oncology clinic with new onset GI symptoms since radiotherapy persisting or starting at least six months following radiotherapy treatment for prostate cancer; any carers or companions of patients as above were eligible for qualitative interviewing study; aged 18 or over; able to undertake an interview in English without the need for translation. Participants were excluded if they had any factors that affected communication or comprehension, lacked capacity to consent or had a known cancer recurrence. To benefit the maximum number of men, patients were recruited until service capacity was reached. All eligible patients were given verbal information and a written information pack about the study at the time of referral to the service. Men who agreed to enter the study were asked to return a signed expression of interest form to the research team. Signed, written consent was taken at least 24 hours after the patient had been given the information pack by a trained member of the research team. Patients were managed within the Enhanced GI Service for 12 months, with an expected average of four appointments. Consenting men completed ALERT-B and the GSRS. If scores were positive due to new onset GI symptoms since radiotherapy, they were eligible for the main EAGLE study and offered referral to the local Enhanced GI Service. These patients and any support-giver or companions (as we are referring to as support-givers) were eligible for the qualitative interviewing study.

## Data collection from the main EAGEL study

Data collection took place at baseline, six months (± 2 months) and 12 months (± 2 months) using case report forms and proformas for QOL assessments to capture acceptability and effectiveness of the intervention. These included: Bowel specific Health related Quality of Life (HRQoL); Global HRQoL; Prostate specific HRQoL; Expanded Prostate cancer Index Composite (EPIC); Diarrhoea scale (GSRS); European Organisation for Research and Treatment of Cancer (EORTC); Patient, support-giver and staff experience; Healthcare resource utilisation (HRU); Cost and acceptability of staff training [24]. The attitudes of patients, support-givers and healthcare professionals towards the new service, as well as the acceptability of staff training, was qualitatively analysed. Details for the statistical analysis including HRU and cost effectiveness and are detailed in our trial protocol paper, and the qualitative analysis are outline below and included in the published protocol paper [24].

## Statistical analysis

Descriptive statistics effect sizes, graphical methods and statistical inferential tests were used to explore participant and treatment characteristics. Cronbach's alpha, item-scale correlation coefficients and intra-class correlation coefficients were used to determine the reliability of outcome measures. Cross-measure correlations were used to measure the validity of outcomes. Cluster analysis was used to measure the heterogeneity in outcome measures across the sites; multilevel techniques or cluster weights were used when data were heterogenous. Standard methods for comparing paired or repeated-measures data were employed in order test if improvements are statistically significant at the 5% level (e.g., paired tests and repeated-measures analysis of variance (ANOVA) using parametric or non-parametric methods, as appropriate). Improvement in outcomes were then reported by comparing the 6 and 12-month follow-up measures to the baseline measurements. Effect sizes such as standardised means from pre- to post-treatment were used to characterise these improvements.

## Health resource utilisation and cost effectiveness analysis

A cost-effectiveness analysis comparing participants not receiving the service was undertaken from an NHS perspective. Total costs (intervention plus/minus any subsequent differences in NHS costs) were assessed against effects in terms of Quality Adjusted Life Years (QALY) based on the EQ-5D-5L. EQ-5D-5L scores are converted to health utilities (1 = perfect health, 0 = equivalent to dead) using a tariff provided by the EuroQol group derived from UK social preference surveys. Resulting utilities are combined with survival data (unlikely to be affected by service) and expressed in QALYs. The estimated incremental cost per QALY from the service can be compared with the willingness to pay threshold of £20,000 - £30,000 per extra QALY currently used by NICE to determine whether an intervention is 'cost effective' and hence recommended for use in the NHS. A secondary analysis was also undertaken from a wider perspective including out-of- pocket expenses borne by the participants and productivity costs associated with time off work.

## Qualitative analysis

The attitudes of patients, support-givers and healthcare professionals towards the new service, as well as the acceptability of staff training, was qualitatively analysed using semi-structured interviews with healthcare professionals and patients, including their companions, at baseline, six and twelve months [24]. Interviews were audio-recorded, transcribed verbatim and anonymised. QSR NVivo 11 qualitative software programme for efficient data management was used to analyse the data adopting Framework Analysis, a pragmatic qualitative approach to qualitative analysis [30–32]. The structure of the Framework Analysis involved five interrelating analysis techniques with associated methods of data ordering: i) familiarisation, where the researcher becomes immersed in the data; ii) developing a thematic framework, where a hierarchical thematic framework is developed to classify and organise data into key themes, concepts and categories; iii) indexing, where the framework is applied to the original data transcripts and coded accordingly; iv) charting, where each theme is charted using a table or matrix using summaries of the data; v) mapping and interpretation, where the charts and data are examined for patterns and connections. All interview data was initially analysed by the qualitative researcher (EB). Ten per cent of transcripts were co-coded by a senior researcher (SS). The interview technique is an iterative process with each interview building on the recognition of themes of interest from the previous. To ensure the findings were grouped into data sets, the researcher received ongoing supervision and had discussions with senior researchers (SS/AN) who supported reflection on their interpretation. The anonymised data were represented by selected extracts in a narrative format with a thematic structure. The results were discussed with data extracts used in support of claims made.

## Patient and public involvement

Patients and public were involved in the project from the initial funding application to study completion. Two research partners (Mrs Susan Campbell and Father Raymond O'Shea) who both had personal experience of the impact of GI pelvic radiotherapy and the impact this has on family and friends, were included in the Study Management Group and directly involved with the drafting of appropriate participant-facing study documentation (including information sheets and consent forms, interview schedules and questionnaires). They were also involved in the discussion and interpretation of the interim and final results and contributed to the funders' reports and final internal reports, along with dissemination, including academic publications.

## Results

### Study population

The service was successfully implemented in three sites, two of which were led by a clinical nurse specialist (CNS) and the third was doctor-led (either by a research fellow or medical speciality registrar (StR). Data was collected between June 2015 and June 2016. A total of 339 men were screened of which 27% (n = 91) were assessed as having at least one positive symptom in the ALERT-B tool and referred to gastroenterology; 58 participants accepted the referral. Thirty-six men attended the 6 month follow-up and 23 men attended the 12 month follow-up. Twenty-three men who had positive symptoms on screening, declined the referral, viewing their symptoms to be trivial, had long-standing symptoms, were already fully investigated or did not want further investigations at that stage (See Fig 3). Table 1 presents the baseline data from the participants referred to the gastroenterology service gastroenterology service showing age, and across the three sites, time to baseline from radiotherapy, treatment techniques, dose fractionation, ADT, T stage, N stage and Gleason scores. The range of eventual diagnoses made included: radiation proctopathy (n = 18); colorectal adenomas (n = 5); non-prostate cancers (n = 3); bile acid malabsorption (n = 15); fructose or lactose intolerance and/or small intestinal bacterial overgrowth (n = 20); vitamin B12/D deficiency (n = 20); this was consistent with similar studies [33, 34].

### Statistical analysis

The EAGLE study was ultimately underpowered for its main analyses due to low recruitment and attrition as described above. However, although results must be interpreted with caution

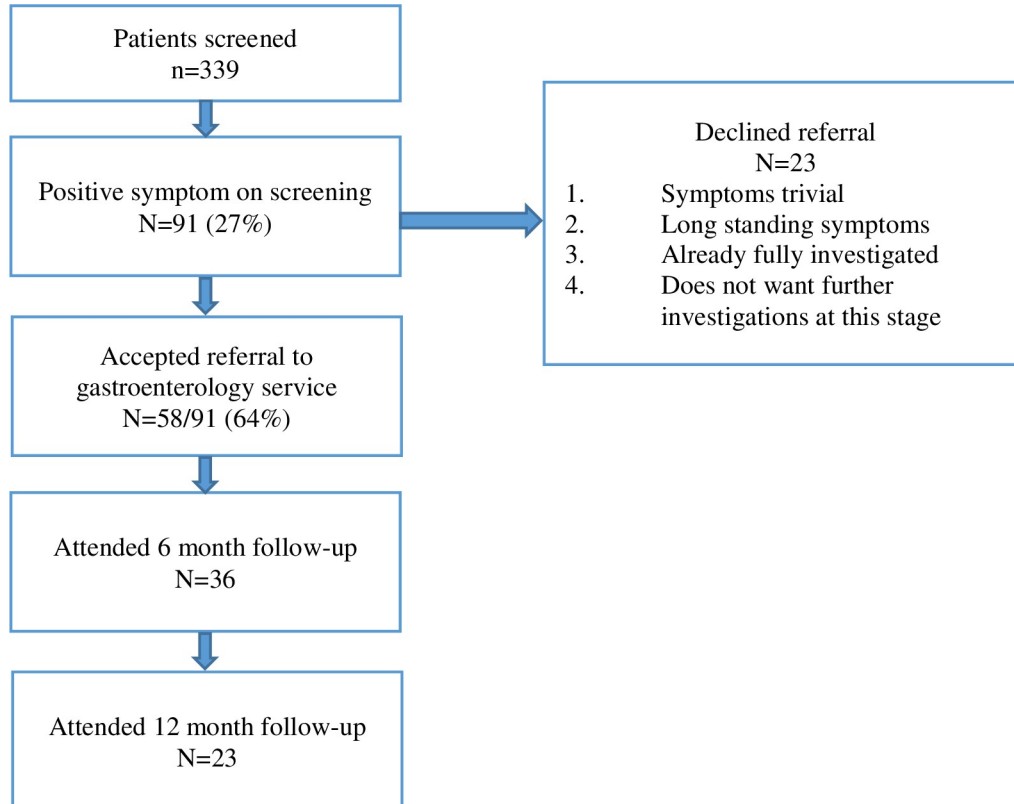

**Fig 3. Centres and patients.** All three centres set-up a specialist PRD service; two with the nurse-led model (Cardiff and Brighton) and one with a consultant led-model (Sheffield).

**Table 1. Baseline data from participants referred to the gastroenterology service.**

| Age years (median) | | 68.3 years (N = 236) | | | |
|---|---|---|---|---|---|
| **Site** | | **Cardiff** | **Sheffield** | **Brighton** | **All** |
| **Time to baseline from radiotherapy** | 0–3 months | n = 30 | n = 17 | n = 3 | N = 50 |
| | >3–9 months | n = 0 | n = 3 | n = 0 | N = 3 |
| | >9–15 months | n = 2 | n = 1 | n = 0 | N = 3 |
| | >15 months | n = 0 | n = 0 | n = 0 | N = 0 |
| **Treatment technique** | EBRT* Only | n = 31 | n = 22 | n = 3 | N = 56 |
| | Combined (EBRT* and Brachytherapy) | n = 1 | n = 0 | n = 0 | N = 1 |
| | Prostate or Prostate bed | n = 26 | n = 20 | n = 3 | N = 49 |
| | Prostate and Pelvic Nodes | n = 5 | n = 2 | n = 0 | N = 7 |
| | Other | n = 1 | n = 0 | n = 0 | N = 1 |
| Dose fractionation** | 46Gray (Gy)/23fractions (fr) | n = 1 | n = 1 | n = 0 | N = 2 |
| | 52.5Gy/20fr | n = 0 | n = 2 | n = 0 | N = 2 |
| | 55Gy/20fr | n = 0 | n = 1 | n = 0 | N = 1 |
| | 57Gy/19fr | n = 0 | n = 0 | n = 1 | N = 1 |
| | 60Gy/20fr | n = 12 | n = 0 | n = 0 | N = 12 |
| | 70Gy/35fr | n = 0 | n = 11 | n = 0 | N = 11 |
| | 74Gy/37fr | n = 9 | n = 7 | n = 2 | N = 18 |
| ADT** | Yes | n = 17 | n = 12 | n = 3 | N = 32 |
| | No | n = 5 | n = 7 | n = 0 | N = 12 |
| **T stage** | T1 | n = 3 | n = 3 | n = 0 | N = 6 |
| | T2 | n = 12 | n = 9 | n = 1 | N = 22 |
| | T3 | n = 6 | n = 9 | n = 2 | N = 17 |
| **N stage** | N0 | n = 19 | n = 20 | n = 3 | N = 42 |
| | N1 | n = 2 | n = 1 | n = 0 | N = 3 |
| **Gleason score** | 3+3 | n = 1 | n = 0 | n = 0 | N = 1 |
| | 3+4 | n = 13 | n = 14 | n = 1 | N = 28 |
| | 4+3 | n = 4 | n = 8 | n = 1 | N = 13 |
| | Higher | n = 1 | n = 0 | n = 1 | N = 2 |

*EBRT = External Beam Radiation Therapy

**ADT = androgen deprivation therapy

**The fractionations schedules used represent standard fractatationation schedules in use of men with prostate cancer in the UK at the time of this study. 60Gy/20 fr is the most common schedule following the CHHiP trial's results,[7] but lower does may be used for less fit men or in the post-operative setting.

due to attrition, Fig 4 demonstrates a consistent picture that the proportion of patients with bowel and urinary symptoms strongly reduced over time. In addition, we observed an increase in quality of life, a reduction in urinary related symptoms, an increase in sexual activity, and/or sexual function between 6 and 12 months; the strongest improvements occurred in the bowel domain. See Table 2 for selected results as the standardized difference from baseline.

## Health economic analysis and costs

HRU modelling was limited by lack of power and attrition. The total costs associated with the intervention, including primary and secondary care contacts, investigations and treatment, and medications averages at £2,243.7 per patient. Staffing and training costs per patient visit were measured and valued at two sites: Nurse-led service (Cardiff) = £67.02; doctor-led service (Sheffield) = £95.25.

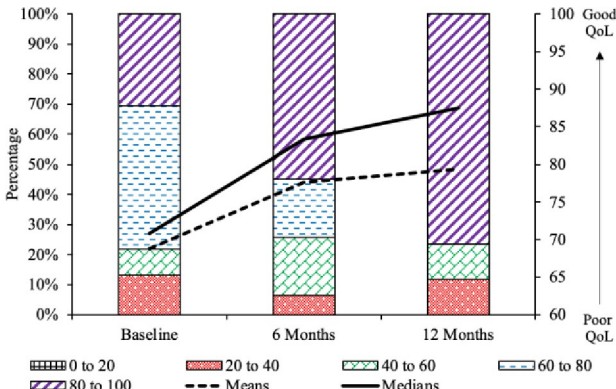

*(A) EPIC Bowel subscale*

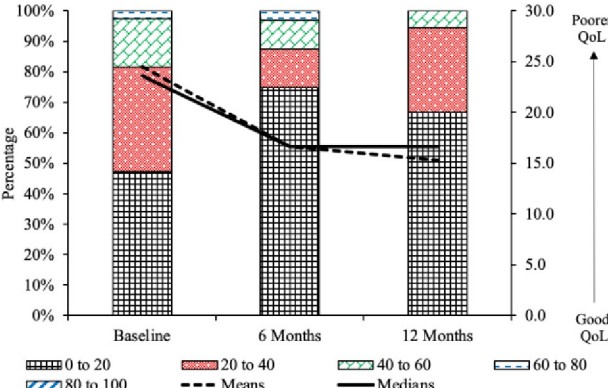

*(B) EORTC Bowel subscale*

**Fig 4. Results for the EPIC bowel subscale and the EORTC QLQ PR25 bowel symptoms scale. 4a) EPIC Bowel subscale.** The results are presented on a scale of 0 (= Extremely Poor QoL) to 100 (= Extremely Good QoL). A statistically significant (6 months versus baseline, P = 0.005; 12 months versus baseline, P = 0.013) increase occurs in bowel QoL (i.e., a reduction in symptoms). The left-hand axis indicates percentages for the stacked bar chart, whereas means and medians are interpreted via the scale on the shown right-hand axis. As the size of the rectangle for 80 to 100 increased substantially for 6 months and 12 months compared to baseline, this indicates that quality of life increased with time (as expected) as EPIC measures QoL on a positive scale. This is also reflected in the mean and median QoL scores (w.r.t. the right-hand axis), which increased in magnitude with time also. **4b) EORTC Bowel subscale.** The results are presented on a scale of 0 (= Extremely Good QoL) to 100 (= Extremely Poor QoL). A statistically significant (6 months versus baseline, P = 0.021; 12 months versus baseline, P = 0.009) increase occurs in bowel QoL (i.e., a reduction in symptoms). The left-hand axis indicates percentages for the stacked bar chart, whereas means and medians are interpreted via the scale on the shown right-hand axis. As the size of the rectangle for 0 to 20 increased substantially for 6 months and 12 months compared to baseline, this indicates that quality of life increased with time (as expected) as EORTC measures QoL on a negative scale. This is also reflected in the mean and median QoL scores (w.r.t. the right-hand axis), which decreased in magnitude with time also.

## Qualitative analysis

Forty-eight interviews were held with a total of 51 interviewees, across the three time points (see Table 3). Four major themes constituted the framework for all interviews: *Barriers and facilitators*; *Capacity*; *Expertise*; *Making a Difference*. These data are to be presented in full in a later paper.

**Table 2. Selected results as the standardized difference from baseline.**

| Scale | Measure: | (mean 6 ms–mean baseline) ÷ SD baseline | (mean 12 ms–mean baseline) ÷ SD baseline |
|---|---|---|---|
| EPIC | Bowel scale | **0.44**** | **0.53*** |
| GSRS | Diarrhoea scale | 0.04 | **0.90*** |
| EORTC | Bowel scale | **0.46*** | **0.55**** |
| EPIC | Urinary Irritative / Obstructive scale | 0.11 | **0.37*** |
| EPIC | Urinary Incontinence | -0.07 | 0.22 |
| EORTC | Urinary scale | 0.09 | **0.46*** |
| EQ-5D-5L | Pain | 0.03 | -0.06 |
| EPIC | Sexual Scale | 0.16 | 0.23* |
| EORTC | Sexual activity | -0.04 | **0.55** |
| EPIC | Hormonal Scale | 0.14* | **0.37** |
| EORTC | Hormones Treatment Related | 0.19 | 0.29 |
| EQ-5D-5L | Anxiety / Depression | -0.12 | 0.07 |
| EORTC | Global QoL | -0.09 | 0.29 |

Note: negative scores indicate a decrease in QoL/symptoms whereas a positive score indicates an increase in QoL/symptoms.

Results for the standardized difference at 6 and 12 months compared to baseline measurements for (selected) subscale scores for the EPIC, EORTC, GSRS and EQ-5D-5L questionnaires. Results are presented such that negative scores indicate a decrease in QoL / increase in symptoms whereas a positive score indicates an increase in QoL / decrease in symptoms for all scales. (Wilcoxon Signed-Rank test: *$P < 0.05$; **$P < 0.01$; ***$P < 0.001$).

**Barriers and facilitators.** Patients felt the service would benefit men after radiotherapy for prostate cancer and were willing to join both the gastroenterology service and the research study to evaluate the service; there were some participants though, who did not differentiate between the two. Barriers reflected the research process itself, reporting concerns around the number of questionnaires they were asked to complete, along with external barriers (such as car parking at NHS centres).

*"he's started to get the investigations done. . ..We have already picked up, something–you know, nothing awful–but I'm thinking, 'Oh, my word'. I just hope his other investigations are going to show loads of problems that he can do something about. . .because he's clearly pinning all his hopes on this service now. . ." (Clinical Nurse Specialist HCP1001_12 months)*

**Capacity.** Patients and HCPs were supportive of the gastroenterology service and expressed hope that the service would continue once the research funding had ended, with some concerns about the sustainability of the service.

*"um when we we've actually got to the point where my post is funded I think it'll be really useful to see if I can spend time with other centres just seeing how they've used with how they are actually putting this the clinic part of it into practice so the practical issues I think would be really helpful" (Clinical Nurse Specialist HP3005_Baseline)*

**Table 3. Breakdown of participants interviewed for the qualitative analysis, over three time points.**

| Participants | Baseline | 6 Months | 12 Months | Total |
|---|---|---|---|---|
| **Gastroenterology** | n = 8 | n = 5 | n = 2 | N = 15 |
| **Oncology** | n = 9 | n = 8 | n = 0 | N = 17 |
| **Patients** | n = 9 | n = 5 | n = 2 | N = 16 |
| TOTAL | N = 26 | N = 18 | N = 4 | N = 48 |

**Expertise.**   Both patients and HCPs welcomed and recognised the expertise of the specialist nurse role; patients had no specific preference as to whether the specialist nurse or a consultant ran the service, whilst for HCPs the specialist nurse role freed up consultant time, so long as they had that level of expertise. HCPs reported good collaboration, attending MDT meetings and providing and/or receiving regular updates on the study, with most following the investigations as the algorithm stated.

*"I mean the nurse I initially saw when I first diagnosed with this cancer they she knew as soon as she inspects you, you know had a look at me she knew and she called (clears throat) she called the doctor straight away I mean she knew its no doubt about it she knew what she was talking about so yeah excellent" (Patient 1A_Baseline)*

**Making a difference.**   Patients reported that referral to the new gastroenterology service helped identify and improve their symptoms. The information given to them to manage their symptoms was found to be useful and effective, making a positive difference for them; advice and support from the dietician in particular were found to make a positive improvement to patients' symptoms. Patients felt that their successful cancer treatment was the most important thing for them, and as a result, they had learned to adapt and self-manage their bowel symptoms themselves; however, referral to the new gastroenterology service helped identify and improve their symptoms. The oncology professionals became more aware of a wider cohort of patients with unmet needs due to pelvic radiotherapy, and who may benefit from access to the service, as a result of the study.

*"...Yes. A very definitely positive change. I mean, because it was getting to the stage where I was (pause) anxious not to, to eat, because I was afraid of, of bringing up the food ... So, I'm eating much better now and I'm much...and my appetite is back and, I'm not afraid to, to eat, if you know what I mean." (Patient 1F_12 month)*

## Discussion

The EAGLE study allowed us to successfully implement a functional service in three NHS centres, facilitated using the ALERT-B tool to highlight symptoms which warranted investigation. We were able to demonstrate a positive impact on the patients who were referred to the service and promoted the development of an interdisciplinary service, with the potential to be of reasonable cost, particularly when using a nurse-led model.

Participants referred from both primary and secondary care (including patients from urology oncology and colorectal surgery) to the Enhanced GI Services—comprising gastroenterologists, CNSs, dieticians and supportive care—had a wide range of symptoms which had an impact on their quality of life and had multiple underlying diagnoses, consistent with a previous national survey [9]. Bowel and urinary symptoms were particularly prevalent in the EAGLE participants; these symptoms improved at both 6 and 12 months, along with a reported increase in sexual activity and functioning. The service, and in particular the dietician support, helped participants manage their symptoms, reporting a positive difference to their quality of life. There is evidence in the literature that men can regret their choice of cancer treatment [13], The men in the EAGLE study did not explicitly state this, rather commented that their symptoms were something they thought they would have to learn to live with. It is possible that this may be one of the reasons why the number of positive symptoms picked up by the ALERT-B tool and of accepted referrals to the service was lower than we anticipated; the well-being of the men is demonstrated in the improvement in symptoms throughout the

duration of the study. However, we cannot be certain on that involvement in the study was the reason. However, we did not collect any data on data on treatment regret and therefore unable comment on associated experience of RT side-effects with treatment regret. From the interviews with the HCPs, we know that men with positive symptoms at screening who declined onward referral for assessment and investigations described their symptoms as trivial. Importantly, EAGLE established specialist GI services, facilitated an increase in awareness of late effects for those who may have unmet needs caused by PRD and put in place robust referral pathways [35].

Exploratory economic analysis identified the nurse-led service to be associated with lower costs than the doctor-led service, with total costs of approximately £2247 per patient. This was also found in the economic analysis of the ORBIT study [36], which also showed that the improvement in symptoms was similar between the two staff groups [37]. A more pragmatic analysis of patients with both urological and gynecological cancer patients reported a mean cost of £1563 per patient [38]. Patients were not concerned as to who led the service, so long as they were qualified to undertake that role, whilst consultants recognized the specialist nurse's expertise and found the nurse-led model freed up consultants' time. There were some concerns as to whether the service would continue after the research funding came to an end, despite efforts made with commissioning bodies. Notably, other centres trialling the algorithm have also found it to effectively improve patients' symptoms at modest cost [39].

## Strengths and limitations

Overall, the EAGLE study made a notable difference for the men who were referred to the new gastroenterology service, reflecting the results of two patterns of care surveys in the literature [9, 34]. Actively implementing these elements in the GI service and evaluating these, and in particularly subjectively, providing us with evidence of the benefit the service provided, for both patients and clinicians. We note that the data is limited by the absence of follow-up longer than 15 months and acknowledge that most relevant late side effects tend to arise more than 1 year post primary radiotherapy [6]. It is important to consider the difference made to the participants who took part and resulted in a quicker implementation of the service. While it is not possible to demonstrate significance in the data from a statistical or economical perspective, the combination of these data provides enough confidence for the services to continue. The HCPs involved in the study reported that they hoped for the services to continue, demonstrating their perceived benefits for men. Indeed, the service in one of our sites has continued within the Gastroenterology department in the local hospital, led by a Gastroenterology CNS [39].

## Conclusions and recommendations

The EAGLE study established new, pragmatic services which, along with the use of the ALERT-B tool, made a positive difference for men with PRD after prostate radiotherapy, and has potential to be of reasonable cost. We believe our results have relevance for radiotherapy services managing patients having radiotherapy for other pelvic malignancies. Our results suggest long-term screening with tools such as ALERT-B for men who have had prostate radiotherapy should be standard of care, whatever follow-up model is offered.

We recommend a nurse-led service, based in gastroenterology with dietician support, and with specialist team members in gastroenterology linking with a late effects clinic based in oncology to ensure potential unmet needs can be picked up timely.

## Supporting information

**S1 Data.**
(XLSX)

**S2 Data.**
(XLSX)

**S1 File.**
(SAV)

**S2 File.**
(SAV)

**S3 File.**
(SAV)

**S4 File.**
(SAV)

**S5 File.**
(SAV)

**S6 File.**
(SAV)

**S7 File.**
(SAV)

## Acknowledgments

We are indebted to the late Rev. O'Shea for his support and involvement in the EAGLE study as a research partner.

## Author Contributions

**Conceptualization:** John N. Staffurth, Raymond O'Shea, Annmarie Nelson.

**Formal analysis:** Stephanie Sivell, Elin Baddeley, Damian J. J. Farnell, Sara Pickett, Sophia Taylor.

**Funding acquisition:** John N. Staffurth, Annmarie Nelson.

**Methodology:** John N. Staffurth, Annmarie Nelson.

**Supervision:** John N. Staffurth, Annmarie Nelson.

**Writing – original draft:** John N. Staffurth, Stephanie Sivell.

**Writing – review & editing:** Elin Baddeley, Sam Ahmedzai, H. Jervoise Andreyev, Susan Campbell, Damian J. J. Farnell, Catherine Ferguson, John Green, Ann Muls, Raymond O'Shea, Sara Pickett, Lesley Smith, Sophia Taylor, Annmarie Nelson.

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
