## [Decision Letter · Decision Letter 0]

4 Dec 2023

PONE-D-23-25824The impact of specialised gastroenterology services for pelvic radiation disease (PRD): results from the prospective multi-centre EAGLE studyPLOS ONE

Dear Dr. Sivell,

Thank you for submitting your manuscript to PLOS ONE. After careful consideration, we feel that it has merit but does not fully meet PLOS ONE’s publication criteria as it currently stands. Therefore, we invite you to submit a revised version of the manuscript that addresses the points raised during the review process.

We look forward to receiving your revised manuscript.

Kind regards,

Bruno Fionda

Academic Editor

PLOS ONE

 [Funding: This study was funded by Prostate Cancer UK’s TrueNTH initiative (Grant Reference No. 250-55).  

AN and SS’ posts were supported by Marie Curie core grant funding, Grant Reference MCCC-FCO-11-C. AN and SS\\ posts are currently supported by Marie Curie, core grant funding, Grant Reference MCCC-FCO-23-C.].  

7. We note that you have stated that you will provide repository information for your data at acceptance. Should your manuscript be accepted for publication, we will hold it until you provide the relevant accession numbers or DOIs necessary to access your data. If you wish to make changes to your Data Availability statement, please describe these changes in your cover letter and we will update your Data Availability statement to reflect the information you provide.

8. Please amend the manuscript submission data (via Edit Submission) to include author Raymond O’Shea. 

9. One of the noted authors is a group [OCCAMS Study Group]. In addition to naming the author group, please list the individual authors and affiliations within this group in the acknowledgments section of your manuscript. Please also indicate clearly a lead author for this group along with a contact email address.

10. Your ethics statement should only appear in the Methods section of your manuscript. If your ethics statement is written in any section besides the Methods, please move it to the Methods section and delete it from any other section. Please ensure that your ethics statement is included in your manuscript, as the ethics statement entered into the online submission form will not be published alongside your manuscript. 

Reviewers' comments:

Reviewer's Responses to Questions

**Comments to the Author**

1. Is the manuscript technically sound, and do the data support the conclusions?

Reviewer #1: Yes

2. Has the statistical analysis been performed appropriately and rigorously? 

Reviewer #1: Yes

3. Have the authors made all data underlying the findings in their manuscript fully available?

Reviewer #1: Yes

4. Is the manuscript presented in an intelligible fashion and written in standard English?

Reviewer #1: Yes

5. Review Comments to the Author

Reviewer #1: This is very interesting study on often underappreciated issue of late radiotherapy side-effect after curative intent radiotherapy. Authors designed referral pattern for patients who experienced late GI symptoms which could be related to radiotherapy. This paper certainly deserve to be published, however, there are few weak points which should be addressed.

I do not see clear definition of the PRD in the introduction, neither definition. How prevalent is PRD? Moreover, radiation proctopathy is not MESH supported terminology, term radiation proctitis is the most accurate one. Which symptoms PRD entails? Please list them as this remained vague in the paper. I would suggest few sentences on etiology of PRD, why is late radiation damage occurring? Spell it out: fibrosis, teleangiektasia, vasculopathy, etc...What was the incidence of more severe GI events, like need for laser coagulation of bleeding rectal teleangiektasia, were there any patients with fistulas?

Can you associate experience of RT side-effects with treatment regret? Did you collect data on that?

I do not see ALERT-B sample questionnaire, actually people in RT world are not familiar with this instrument, presenting this questionnaire in appendix would increase papers clarity and integrity.

How this study improved the well-being of 47 men with gastrointestinal late effects? This remained obscured?

I do not see sample size calculation, is this ad-hoc sample of how many you were able to collect or you did pre-study calculation. If yes, what was your assumed incidence of RT-related GI side-effects. Please elaborate.

Please explain sharp drop-off from 6-months to 12-months follow-up point (Figure 2).

I have noticed weird fractionation schedule used: 55 Gy/20 fr, 52.5 Gy/20 fr. These are not typical regimens for prostate cancer but are common for bladder cancer. Please check or explain.

Important limitation of this paper is absence of follow-up longer than 15 months as the majority of the most relevant late side effects occur 1 year after primary radiotherapy. Please list this limitation.

Figure 3a and 3b: "The left-hand axis indicates percentages for the stacked bar chart", ok, but explain what squares with 0-20, 20-40, 40-60 etc represent.

Label of Table 3 is not easy comprehensible. Please reformulate.

Authors mention peer reviewed algorithm for pelvic radiation disease but does not explicitly state details of this algorithm. Is this published work?

6. PLOS authors have the option to publish the peer review history of their article (what does this mean?). If published, this will include your full peer review and any attached files.

Reviewer #1: **Yes: **Jure Murgic

---

## [Author Response · Author response to Decision Letter 0]

1 Mar 2024

Comments/questions from the Editors and Peer Reviewers Response

https://journals.plos.org/plosone/s/file?id=ba62/PLOSOne_formatting_sample_title_authors_affiliations.pdf Complete

2. Please provide additional details regarding participant consent.. In the ethics statement in the Methods and online submission information, please ensure that you have specified what type you obtained (for instance, written or verbal, and if verbal, how it was documented and witnessed). If your study included minors, state whether you obtained consent from parents or guardians. If the need for consent was waived by the ethics committee, please include this information.

Once you have amended this/these statement(s) in the Methods section of the manuscript, please add the same text to the “Ethics Statement” field of the submission form (via “Edit Submission”). We obtained signed, written consent. We have added this to the methods section and will add the same text in the Ethics statement field in the submission form. See line 174-178:

“All eligible patients were given verbal information and a written information pack about the study at the time of referral to the service. Men who agreed to enter the study were asked to return a signed expression of interest form to the research team. Signed, written consent was taken at least 24 hours after the patient had been given the information pack by a trained member of the research team”

 We have attached the minimal statistical data via SPSS and EXCEL files as supplementary information.

When you resubmit, please ensure that you provide the correct grant numbers for the awards you received for your study in the ‘Funding Information’ section. Our apologies; we have inadvertently included information from a separate project.

The relevant and accurate funding is included within the Acknowledgments on lines 419-422:

Funding: This study was funded by Prostate Cancer UK’s TrueNTH initiative (Grant Reference No. 250-55). The funders had no role in study design, data collection and analysis, decision to publish, or preparation of the manuscript. AN and SS’ posts were supported by Marie Curie core grant funding, Grant Reference MCCC-FCO-11-C. AN and SS’ posts are currently supported by Marie Curie, core grant funding, Grant Reference MCCC-FCO-23-C. 

We will ensure this matches the information on the submission portal as well.

 [Funding: This study was funded by Prostate Cancer UK’s TrueNTH initiative (Grant Reference No. 250-55). 

AN and SS’ posts were supported by Marie Curie core grant funding, Grant Reference MCCC-FCO-11-C. AN and SS\\ posts are currently supported by Marie Curie, core grant funding, Grant Reference MCCC-FCO-23-C.]. 

Please include this amended Role of Funder statement in your cover letter; we will change the online submission form on your behalf. We can confirm that the funders had “no role” and have added that statement to funders section – as above in Q4. 

We will update your Data Availability statement to reflect the information you provide in your cover letter. As referred to in Q3, we have attached the minimal statistical data via SPSS and EXCEL files as supplementary information.

7. We note that you have stated that you will provide repository information for your data at acceptance. Should your manuscript be accepted for publication, we will hold it until you provide the relevant accession numbers or DOIs necessary to access your data. If you wish to make changes to your Data Availability statement, please describe these changes in your cover letter and we will update your Data Availability statement to reflect the information you provide.

 As referred to in Q3, we have attached the minimal statistical data via SPSS and EXCEL files as supplementary information.

Please amend the manuscript submission data (via Edit Submission) to include author Raymond O’Shea. We were sorry to report that Rayon O’Shea who was one of the project’s patient representative’s and unfortunately died during the submission of the manuscript. We are therefore not able to add him to the system. We can of course remove as a co-author and maintain the reference in acknowledgement if we are not able to keep his status as a co-author and happy to take the editorial board’s views on this matter.

9. One of the noted authors is a group [OCCAMS Study Group]. In addition to naming the author group, please list the individual authors and affiliations within this group in the acknowledgments section of your manuscript. Please also indicate clearly a lead author for this group along with a contact email address.

 In relation to Q4, we have inadvertently included information from a different project. We have removed this information from the manuscript. 

10. Your ethics statement should only appear in the Methods section of your manuscript. If your ethics statement is written in any section besides the Methods, please move it to the Methods section and delete it from any other section. Please ensure that your ethics statement is included in your manuscript, as the ethics statement entered into the online submission form will not be published alongside your manuscript. 

 We have moved the ethics statement from the Acknowledgements section to the Methods section on Lines 129-130

“Ethical approval was received from the NHS Research Ethics Committee: NRES Committee North West-Liverpool East REC (Reference 14/NW/1206).”

11. Please review your reference list to ensure that it is complete and correct. If you have cited papers that have been retracted, please include the rationale for doing so in the manuscript text, or remove these references and replace them with relevant current references. Any changes to the reference list should be mentioned in the rebuttal letter that accompanies your revised manuscript. If you need to cite a retracted article, indicate the article’s retracted status in the References list and also include a citation and full reference for the retraction notice. The references have been updated since revising the manuscript.

I do not see clear definition of the PRD in the introduction, neither definition. How prevalent is PRD? Moreover, radiation proctopathy is not MESH supported terminology, term radiation proctitis is the most accurate one. Which symptoms PRD entails? Please list them as this remained vague in the paper. I would suggest few sentences on etiology of PRD, why is late radiation damage occurring? Spell it out: fibrosis, teleangiektasia, vasculopathy, etc...What was the incidence of more severe GI events, like need for laser coagulation of bleeding rectal teleangiektasia, were there any patients with fistulas?

 We have added and updated the Introduction section, see lines 91-120:

Late effects of PRD can affect multiple domains: broadly separated into gastrointestinal, genitourinary, sexual, endocrine, skeletal, psychological and fatigue. While there is some uncertainty over the true prevalence of late effects.7,8 It is important to recognise that these effects on different domains may co-exist in the same patient, and are linked and can that PRD can impact on a man’s psychological and social wellbeing, including feelings of regret at their choice of cancer treatment.9-12 Oncologists often do not seek information on late effects focusing on symptoms of cancer recurrence.4 Patients, viewing them as expected consequences of therapy, may be too embarrassed to mention them, or grateful for being cured, may not mention their symptoms.13 Many different reporting systems and tools are used to collect toxicity and quality of life data in both research and routine practice.7,8,14 There is no agreed definition of pelvic radiation disease and its prevalence depends on radiation delivered (dose, volume and fractionationation), patient factors and follow-up, including extent of investigations for symptoms consistent with PRD.15 Therefore, the prevalence of grade two or higher late gastrointestinal and genitourinary effects after modern radiotherapy for localised prostate cancer is usually <3%,16 whereas the prevalence of grade 3 or higher toxicity maybe as high as 10% after wide-field radiotherapy for cervical cancer or in the peri-operative setting for rectal cancer17,18 prevalence studies may report as high as 40% of patients with long-term bowel dysfunction after such wide-field radiotherapy.19

Acute radiation effects are caused by damage to rapidly proliferating tissues such as gut mucosa.20 The pathophysiology of PRD is more complicated and symptoms may be caused by structural or functional changes. Some patients will have a persistent unchecked inflammatory response with an initial acute inflammatory response complicated by endothelial cell and microvascular damage; this sustains tissue hypoxia, dysregulates wound healing and can lead to fibrosis. Hypoxia leads to abnormal neovascularisation i.e. telangiectatisia. Direct damage can also occur to larger vessels and nerves; often in tissues that have prior surgery as part of the initial cancer management. Fibrosis within the gastrointestinal tract can alter transit time, impair motility, affect nutrient absorption, alter the microbiome and cause strictures, infections and even fistulae.19 These physiological changes may lead to over 20 different gastrointestinal or nutritional diagnoses and 12 different symptom.21 Equivalent changes may affect other domains, although are less well studied.22

Can you associate experience of RT side-effects with treatment regret? Did you collect data on that? We did not collect any data on data on treatment regret and therefore unable comment on associated experience of RT side-effects with treatment regret. We have included this in the Discussion in lines: 346-354

There is evidence in the literature that men can regret their choice of cancer treatment,13 The men in the EAGLE study did not explicitly state this, rather commented that their symptoms were something they thought they would have to learn to live with. It is possible that this may be one of the reasons why the number of positive symptoms picked up by the ALERT-B tool and of accepted referrals to the service was lower than we anticipated; the well-being of the men is demonstrated in the improvement in symptoms throughout the duration of the study. However, we cannot be certain on that involvement in the study was the reason. However, we did not collect any data on data on treatment regret and therefore unable comment on associated experience of RT side-effects with treatment regret. 

I do not see ALERT-B sample questionnaire, actually people in RT world are not familiar with this instrument, presenting this questionnaire in appendix would increase papers clarity and integrity.

 We have included the ALERT- B questionnaire in the Supporting Information section and referred to in lines: 135-146. The ALERT-B questionnaire was originally cited here:

Taylor S, Byrne A, Adams R, Turner J, Hanna L, Staffurth J, et al. The Three-item ALERT-B Questionnaire Provides a Validated Screening Tool to Detect Chronic Gastrointestinal Symptoms after Pelvic Radiotherapy in Cancer Survivors. Clin Oncol (R Coll Radiol). 2016;28(10):e139-47. doi: 10.1016/j.clon.2016.06.004

How this study improved the well-being of 47 men with gastrointestinal late effects? This remained obscured?

 The well-being of the patients is demonstrated in the improvement in symptoms throughout the duration of the study. However, we cannot be certain on that involvement in the study was the reason. We have noted this in the Discussion, lines 346-354

There is evidence in the literature that men can regret their choice of cancer treatment,13 The men in the EAGLE study did not explicitly state this, rather commented that their symptoms were something they thought they would have to learn to live with. It is possible that this may be one of the reasons why the number of positive symptoms picked up by the ALERT-B tool and of accepted referrals to the service was lower than we anticipated; the well-being of the men is demonstrated in the improvement in symptoms throughout the duration of the study. However, we cannot be certain on that involvement in the study was the reason. However, we did not collect any data on data on treatment regret and therefore unable comment on associated experience of RT side-effects with treatment regret. 

I do not see sample size calculation, is this ad-hoc sample of how many you were able to collect or you did pre-study calculation. If yes, what was your assumed incidence of RT-related GI side-effects. Please elaborate.

 We did not have a sample size calculation; the purpose of this was to implement a service and to observe the numbers of uptake to the services.

Please explain sharp drop-off from 6-months to 12-months follow-up point (Figure 2).

 In the Discussion, lines 343-350, we have discussed the potential reasons there was a drop-off between 6-months and 12-months.

“Bowel and urinary symptoms were particularly prevalent in the EAGLE participants; these symptoms improved at both 6 and 12 months, along with a reported increase in sexual activity and functioning. The service, and in particular the dietician support, helped participants manage their symptoms, reporting a positive difference to their quality of life. There is evidence in the literature th

---

## [Editor Report · Decision Letter 1]

24 Apr 2024

The impact of specialised gastroenterology services for pelvic radiation disease (PRD): results from the prospective multi-centre EAGLE study

PONE-D-23-25824R1

Dear Dr. Sivell,

We’re pleased to inform you that your manuscript has been judged scientifically suitable for publication and will be formally accepted for publication once it meets all outstanding technical requirements.

Kind regards,

Bruno Fionda

Academic Editor

PLOS ONE
---

## [Editor Report · Acceptance letter]

23 Oct 2024

PONE-D-23-25824R1 

PLOS ONE

Dear Dr. Sivell, 

I'm pleased to inform you that your manuscript has been deemed suitable for publication in PLOS ONE. Congratulations! Your manuscript is now being handed over to our production team.

Kind regards, 

on behalf of

Dr. Bruno Fionda 

Academic Editor

PLOS ONE